# The *miR-214-5p*/Lactoferrin/*miR-224-5p*/*ADAM17* Axis Is Involved in Goat Mammary Epithelial Cells’ Immune Regulation

**DOI:** 10.3390/ani13182835

**Published:** 2023-09-06

**Authors:** Shilong Pang, Yuexin Shao, Yan Yu, Kela Sha, Yanting Jiang, Xian Zhang, Yuling Zhong, Huaiping Shi, Weijuan Li

**Affiliations:** 1College of Animal Science and Technology, Northwest A&F University, Yangling 712100, China; pangslxxx@gmail.com (S.P.); shaoyuexin0423@163.com (Y.S.); yuyan991101@163.com (Y.Y.); kelashasklsci@126.com (K.S.); zhangxian1351371@126.com (X.Z.); m17756572398@163.com (Y.Z.); 2Yunnan Academy of Animal Husbandry and Veterinary Sciences, Kunming 650224, China; jiangyanting-2007@163.com

**Keywords:** lactoferrin, *miR-214-5p*, goat mammary epithelial cells, immune regulation

## Abstract

**Simple Summary:**

Lactoferrin (*LF*) is considered to be an important active protein in goat milk, and it plays an important role in immunity. In this study, we tried to find factors affecting the expression of lactoferrin. Furthermore, we further explored the anti-inflammatory effect of *LF* and related molecular mechanisms in order to further understand how lactoferrin affects the immune system and provide a reference for further study and application of lactoferrin in the future.

**Abstract:**

Lactoferrin (*LF*) is believed to be an important active protein in goat milk, which plays an anti-inflammatory role. Although *LF* has been reported to be associated with body health, its exact underlying mechanism remains unclear. Here, we aimed to elucidate the mechanism of this anti-inflammatory effect of *LF* in vitro. We first identified that *miR-214-5p* inhibited the expression of *LF* mRNA and protein in cells through the 3′UTR of *LF* mRNA. We next identified the alterations in miRNA following *LF* overexpression in goat mammary epithelial cells (GEMCs). Overexpression of *LF* significantly increased (*p* < 0.05) *miR-224-5p* expression. We further revealed that transcriptional activation of *ADAM17*, TNF-α, IL-1β, and IL-6 was efficiently decreased (*p* < 0.05) in GMECs treated by *miR-224-5p* mimic. Conversely, knockdown of *miR-224-5p* increased (*p* < 0.05) *ADAM17*, TNF-α, IL-1β, and IL-6 expression. Additionally, TNF-α, IL-1β, and IL-6 expression levels were dramatically decreased in GMECs after administration of si*ADAM17*. Herein, we indicate that the *miR-214-5p*/*LF*/*miR-224-5p*/*ADAM17* axis is involved in the immune regulation of GEMCs.

## 1. Introduction

Mastitis is a common disease in livestock breeding, which can lead to decreased milk yield, decreased milk quality and huge economic losses. Therefore, how to control the occurrence of mastitis has become the focus of attention in animal husbandry production. But there are relatively few studies on dairy goats. In goat mammary epithelial cells, IL-17A was found to promote the secretion of a variety of pro-inflammatory cytokines, chemokines, and defensive proteins. Results indicate that IL-17A may be involved in the occurrence of mastitis in dairy goats and plays a certain regulatory role in the mammary gland immune defense of goats [1]. Studies show that the expression level of antibacterial peptide S100A7 mRNA in the mammary cells of goats has a good correlation with mastitis, which could reflect the severity of mastitis in a certain range. The expression of antibacterial peptide S100A7 in goat mammary epithelial cells can be induced by LPS. LPS may be involved in regulating the expression and secretion of antimicrobial peptide S100A7 through the TLR4/NF-κB signaling pathway [2]. At this time, lactoferrin, which has good anti-inflammatory and immunomodulatory effects and other biologically active functions, has become the object of our study.

Lactoferrin (*LF*) is mainly present in mammalian milk, and its content is high in colostrum and low in mature milk [3,4,5]. As a widely distributed protein in biological secretions, *LF* plays an important role in innate immunity. A large number of studies have shown that *LF* has strong anti-inflammatory activity, which can not only regulate the inflammatory response of epithelial cells infected with intracellular bacteria [6] but also reduce the inflammatory response triggered by the involvement of TLR [7,8]. The mRNA expression levels of IL-6, IL-8, and TNF-α in Caco-2 cells infected with *E. coli* HB101 were decreased after b*LF* (Bovine Lactoferrin) treatment [9]. Experiments have been conducted to treat mice-fed bacterial lipopolysaccharide (LPS) with b*LF* before inducing systemic inflammation. The results showed that the levels of TNFα, IL-6, and IL-10 in mice significantly decreased after pre-administration of *LF* but no significant changes were found in the above inflammatory factors after feeding with LPS. This suggests that *LF* can counteract the inflammatory response induced by LPS [10]. It was also found that *LF* ingested by newborns can promote the establishment of the natural immune system and enhance the function of the immune system [11].

MicroRNAs (miRNAs) are small noncoding RNAs (~22 nucleotides in length) that modulate gene expression by targeting 3′-untranslated regions (3′UTRs) of mRNA transcripts [8]. Early trials found that miR-214 inhibits the expression of *LF* and promotes apoptosis in human mammary epithelial cells by binding to *LF* mRNA [12]. miRNAs can regulate the post-transcriptional silencing of target genes [13]. A single miRNA can simultaneously target hundreds of mRNAs and affect the expression of multiple genes involved in related functional interactions. And this gives us a starting point. We can explore the specific regulatory pathway of *LF* in the immune response of dairy goat mammary epithelial cells by looking for the upstream and downstream miRNAs and the genes regulated by those miRNAs. *ADAM17* is one of the downstream genes that is associated with immunity. In cardiovascular disease, miR-26a-5p reduces apoptosis and the release of inflammatory cytokines, such as TNF-α, IL-1β, and IL-6, by targeting *ADAM17* [14]. miR-708-3p reduces inflammatory responses and damage to cardiomyocytes after acute myocardial infarction by negatively regulating *ADAM17* [15].

Therefore, in the present study, we focused on identifying upstream and downstream differential miRNAs of *LF* related to immune regulation. Based on this, we further investigated the potential molecular mechanism of *LF* involvement in the immune response of mammary epithelial cells in dairy goats. This study provides a theoretical basis for an *LF* regulatory mechanism in enhancing immune ability, paving the way for improving production efficiency in goats.

## 2. Materials and Methods

### 2.1. Ethics Statement

All animal experiments were carried out under the agreement of the Institutional Animal Use and Care Committee (Northwest A&F University, Yangling, China; permit number DK2022008; 28 February 2022). All surgeries were performed so as to minimize suffering. All animals in the present study received humane care according to the Guide for the Care and Use of Experimental Animals of the National Institutes of Health.

### 2.2. Mammary Tissue Collection and Cell Culture

The GMECs we used were isolated from mammary tissue washed with D-Hank’s solution preserved in our laboratory. The mammary tissue was collected from three individual 3-year-old dairy goats in their peak lactation period. The tissue sections were cut into cubes of about 1 mm^3^ and cultured in a petri dish at 37 °C in 5% CO_2_. The medium was configured according to the formula of our laboratory [16]. The culture medium was changed every 2 days until epithelial cells separated from the tissue block. Cells were subsequently digested from the tissue block with 0.25% trypsin–ethylene diamine tetraacetic acid (EDTA) solution. The differential adhesion method was used to purify GMECs by removing fibroblasts, which adhered to culture dishes 30–40 min faster than GMECs. The suspended GMECs and culture medium were transferred to new culture dishes. The purity of the GMECs after threefold purification was >99%. The GMECs obtained from a five-passage purification process were used in subsequent experiments. The culture medium contained basal DMEM/F12 (SH30023, HyClone, Logan, UT, USA) medium, 1 μg/mL hydrocortisone (H0888, Sigma, St. Louis, MO, USA), 5 μg/mL insulin (16634, Sigma), 10 ng/mL epidermal growth factor (PHG0311, Invitrogen, Carlsbad, CA, USA), 100 U/mL penicillin/streptomycin (080092569, Harbin Pharmaceutical Group, Harbin, China), and 10% fetal bovine serum (10099141, Gibco, Gaithersburg, MD, USA).

### 2.3. MiRNA Sequencing

A mirVana miRNA isolation kit (Thermo Fisher Ambion, Austin, TX, USA) was used to isolate small RNA from GMECs cultured in culture medium containing basal medium for 24 h. Illumina ‘s TruSeq small RNA sample preparation kit A was then used to prepare a small RNA barcode library. Additional size selection of miRNA barcodes with a band size of 147 bp was performed on a 6% TBE PAGE gel to reduce tRNA contamination. An Agilent 2100 bioanalyzer was used to determine the mass and concentration of RNA. We used the Illumina MiSeq system at Stanford’s Functional Genomics Center (SFGC) to perform sequencing of barcoded samples, which were pooled at equal molar ratios. R1 single-end sequencing files were processed in Galaxy (https://usegalaxy.org, 9 July 2023) using tools for quality control (groomer), and adaptor clipping and miRanalyzer (https://bioinfo5.ugr.es/miRanalyzer, 9 July 2023) were used to assess miRNA expression. The Norm-Expressed-Mapped data from the miRanalyzer mature unique reads were used for the scatterplot and differential expression analyses. We used half of the minimum value of the condition to replace the zero value for miRNAs with a zero value in only one condition.

### 2.4. Vector Construction

Based on goat *LF* (NC_030829.1) and the *ADAM17* (XM_005687094.3) CDS we designed the clone primers. All plasmids were confirmed through sequencing before a pcDNA3.1 vector was used to clone the coding sequence of goat *LF*/*ADAM17* between the HindIII (Takara) and XhoI (Takara) restriction sites to produce pcDNA3.1-*LF* and pcDNA3.1-*ADAM17*, respectively.

### 2.5. GMECs Cell Transfection

In the first experiment, GMECs were seeded in 48-well plates until 80% confluence before applying treatments. *miR-214-5p* were predicted to target *LF* 3′UTR using the TargetScan database. To overexpress miRNA, we used Lipofectamine RNAiMAX transfection reagent (13778150, Invitrogen), and, for miRNA knockdown, the inhibitor NC, *miR-214-5p* inhibitor, *miR-214-5p* mimic, and mimic NC (100 nM, RiboBio, Guangzhou, China) were separately transfected into cells. A mimic NC and an inhibitor NC were used as controls, respectively.

In the second experiment, the pcDNA3.1-*LF* recombinant vector was used to overexpress the *LF* gene in GMECs. A pcDNA3.1 vector was used as the control.

In the third experiment, GMECs were seeded in 48-well plates until 80% confluence before applying treatments. The inhibitor NC, *miR-224-5p* inhibitor, *miR-224-5p* mimic, and mimic NC (100 nM, RiboBio) were separately transfected into cells. A mimic NC and inhibitor NC were used as controls, respectively.

In the fourth experiment, small interfering RNA (*siADAM17*, GenePharma, Shanghai, China) was used to knock down *ADAM17*. The *siADAM17* sequences are shown in Table 1. siNRA NCs were used as the control. The negative control siRNA (siNC) sequences were as follows: sense: 5′-UUCUCCGAACGUGUCACGUTT-3′, antisense: 5′-ACGUGACACGUUCGGAGAATT-3′. In addition, the pcDNA3.1-*ADAM17* recombinant vector was used to overexpress the *ADAM17* gene in GMECs. A pcDNA3.1 vector was used as the control.

In the last experiment, for co-treatment assays of *miR-224-5p*, cells were transfected with a total of 100 inhibitors or 100 nM siRNAs in each group (the same proportions of the two ingredients).

Lipofectamine 2000 and Lipofectamine RNAiMAX were used for the transfection of pcDNA3.1-*LF*/*ADAM17*, miRNA mimic and inhibitors (100 nM), and siRNA (100 nM) into GMECs, respectively. After 48 h from the protein extraction and RNA transfection, the GMECs were harvested. A mimic NC, inhibitor NC, and siRNA NC pcDNA3.1 were used as controls. All experiments were carried out in triplicate and repeated three times.

### 2.6. RNA Extraction and Quantitative Real-Time PCR (qRT-PCR)

Before applying treatments, GMECs were cultured until 80% confluence in 12-well plates. RNAiso Plus (9109, Takara) was used to extract the total RNA from each sample. The quality of RNA was detected with a NanoDrop 2000 spectrophotometer (ThermoScientific, Waltham, MA, USA). The cDNA was synthesized with a high-capacity cDNA reverse transcription kit (Applied Biosystems, Foster City, CA, USA). The miRcute cDNA First-Strand kit (KR211, Tiangen, Beijing, China) was used to synthesize the mature miRNAs’ first-strand cDNA for quantitative analysis of miRNAs. A MiRcute miRNA q-PCR kit (FP411, Tiangen) was used to amplify the reaction products, and 5S rRNA was used as an internal control for miRNAs. The primers for qRT-PCR of genes are shown in Table 1. The SYBR Green kit (RR820A, TB Green II, PerfectReal Time; Takara) was used to perform quantitative real-time PCR (qRT-PCR). As for the internal control genes, we chose ubiquitously expressed transcript (UXT) and ribosomal protein S9 (RPS9) for their stable expression during the different lactation stages of dairy goats. The relative expression values were normalized to the level of UXT and RPS9 and calculated by the 2^−ΔΔCt^ method. Each sample was analyzed in three biological replicates.

### 2.7. Western Blot

Before applying treatments, GMECs were cultured until 80% confluence in 6-well plates. Radio immunoprecipitation assay (RIPA) buffer (R0010, Solarbio, Beijing, China) containing protease and phosphatase inhibitor cocktail tablets (04693132001 and 04906845001, Roche, Basel, Switzerland) was used to lyse GMECs before cellular protein was harvested. A BCA protein assay kit (23227, Thermo Scientific, Waltham, MA, USA) was used to examine the protein concentration. Sodium dodecyl sulfate–polyacrylamide gel electrophoresis (SDS-PAGE) was used to separate the protein. Transfer of the separated protein onto a PVDF membrane (Roche) was performed by Trans-Blot SD semidry transfer cell (Bio-Rad, Hercules, CA, USA). A quantity of 5% skim milk (232100, BD Biosciences, Franklin Lakes, NJ, USA) was used to block membranes for 2 h. After membranes were incubated, the secondary antibody horseradish peroxidase HRP-conjugated goat anti-rabbit-IgG (CW0103, CW Biotech; Beijing, China; 1:5000) was used for primary antibodies’ LTF. The antibody HRP-conjugated goat anti-mouse-IgG (CW0102, CW Biotech;1:5000) was used for primary antibody antisera preservation in this laboratory. The primary antibody was incubated at 4 °C for 12 h and the secondary antibody was incubated at 37 °C for 2 h. Enhanced chemiluminescence (ECL) Western blot system (1705061, Bio-Rad) was applied to measure signals. Densitometry using ImageJ software (http://imagej.nih.gov/ij/, 9 July 2023) was used to quantify the intensity of indicated bands.

### 2.8. Luciferase Assays

psi-CHECK2 was used as the luciferase vector. The GMECs were inoculated into 24-well plates. When the cell growth fusion degree reached about 70%, the miRNA mimics were co-transfected with the luciferase vector. First, the luciferase plasmid 0.5 ug was transfected into each well, and then the transfection was mixed well, left standing for 20 min, evenly added to each well, and shaken well. The transfection concentration of *miR-214-5p* mimic/NC was 50 nM, and an equal volume of Lipofectamine RNAiMAX transfection reagent was added. DMEM was then supplemented to 50 μL. The *miR-214-5p* inhibitor/NC transfection concentration was 100 nM, and DMEM was added to a 50 μL volume transfection reagent. The transfection compound was blown and mixed well, left for 20 min, and then uniformly added into each well and mixed by shaking. We repeated with three holes for each addition. A 24-well plate was prepared with luciferase vector and transfected, and the transfection concentrations of *miR-214-5p* mimic and mimic were divided into three gradient levels of 50 nM, 100 nM, and 150 nM. The transfections were performed using normal controls as control groups. Using the constructed dual-luciferase vector as a template, two pairs of mutant primers were designed for PCR amplification. The site sequence that binds to the *miR-214-5p* seed region in the *LF* 3’UTR was successfully mutated. The mutant sequence was digested and ligated with the psi-CHECK2 vector to construct *psi-MUT-LF*. The assay was repeated three times, adding three repeat holes to each hole. Transfection lasted for 48 h, and the dual luciferase activity was detected by enzyme labeling and the Promega reporting system.

### 2.9. Statistical Analysis

All experiments were repeated with at least three biological replicates. SPSS 20.0 (IBM, Chicago, IL, USA) was used to perform the statistical analysis. When only two groups were compared, the data were analyzed using Student’s t-test (two-tailed). *p*-values < 0.05 were considered statistically significant (* *p* < 0.05, ** *p* < 0.01). Values are presented as mean ± standard error of the mean (SEM).

## 3. Results

### 3.1. miR-214-5p Target 3′UTR of LF mRNA

The synthetic miRNA mimic/control and its inhibitor/control were diluted according to standards and transfected into GMECs, respectively. The transfected cell miRNA was extracted for quantitative detection. Results showed that the level of *miR-214-5p* in the cells transfected with *miR-214-5p* mimic increased 87 times (*p* < 0.01, Figure 1A) compared with the *miR-214-5p* mimic control group. More critically, compared with the *miR-214-5p* inhibitor control group, the *miR-214-5p* inhibitor attenuated *miR-214-5p* expression (*p* < 0.01, Figure 1A). Subsequently, compared with the *miR-214-5p* mimic control group, the activity of dual luciferase in cells transfected with *miR-214-5p* mimic was significantly decreased (*p* < 0.05, Figure 1B), as the *miR-214-5p* inhibitor had the opposite effect (*p* < 0.05, Figure 1B).

To target the relationship between *miR-214-5p* and *LF*, *psi-MUT-LF* was respectively co-transfected with *miR-214-5p* mimic/control or *miR-214-5p* inhibitor/control in GMECs. The results of applying luciferase assays after transfection with *miR-214-5p* mimic and inhibitor showed no significant change compared with the control (*p > 0.05*, Figure 1C). This indicates that *miR-214-5p* is bound to the 3′UTR of *LF* mRNA. After transfection of the *miR-214-5p* mimic, the intracellular *LF* mRNA level was significantly decreased (*p* < 0.01, Figure 1D). In contrast, *LF* expression increased in the *miR-214-5p* inhibitor group (*p* < 0.01, Figure 1D). These results indicate that *miR-214-5p* can inhibit the expression of *LF* mRNA and protein in cells.

### 3.2. miR-214-5p Decreases the Expression of Inflammatory Factors

The *miR-214-5p* mimic/inhibitor and its respective control were transfected into GMECs to quantitatively detect the expression of related inflammatory factors in cells. The results show that the levels of inflammatory factors IL-6 and IL-8 were significantly increased in cells transfected with *miR-214-5p* mimic (*p* < 0.05) and the levels of anti-inflammatory factor IL-10 were significantly decreased (*p* < 0.05, Figure 1E). The levels of proinflammatory cytokines IL-6 and IL-8 were significantly decreased in *miR-214-5p*-inhibitor-transfected cells (*p* < 0.05) and the levels of anti-inflammatory factor IL-10 were significantly increased (*p* < 0.05, Figure 1F).

### 3.3. LF Decreases the Expression of Inflammatory Factors

The overexpression efficiency of *LF* was tested, and the results show that the expression of *LF* was 78 times greater than in the control group (Figure 2A). We then examined the expressions of inflammatory factors in *LF*-overexpressing GMECs. The results show that, in *LF*-overexpressing GMECs, the levels of proinflammatory factors IL-6 and IL-8 were greatly decreased, and the level of anti-inflammatory factor IL-10 was significantly increased (*p* < 0.05, Figure 2B).

### 3.4. LF Attenuates the Inflammatory Response via Targeting miR-224-5p

We next further elucidate the molecular mechanisms underlying *LF*-induced inflammatory response through miRNA sequencing. miRNA sequencing was performed after *LF* overexpression. According to statistical analysis with R software, 22 differentially expressed miRNAs were identified, including 15 up-regulated miRNAs and 7 down-regulated miRNAs. Among these differential miRNAs, *miR-224-5p* is related to immune regulation (Figure 3A,B). Subsequently, we verified that *miR-224-5p* was up-regulated in the pcDNA3.1-*LF* transfected GMECs through qRT-PCR methods (Figure 3C). To determine whether *miR-224-5p* could affect the expression of inflammation factors, the *miR-224-5p* mimic and inhibitor were transfected into GMECs. The results show that the mRNA levels of TNF-α and IL-8 were greatly decreased in GMECs treated with *miR-224-5p* mimic (*p* < 0.05) (Figure 3D) and the mRNA levels of TNF-α, IL-6, and IL-8 were significantly increased in GMECs treated by *miR-224-5p* (*p* < 0.01, Figure 3E). These results indicate that *LF* decreases the expression of inflammation factors via targeting of *miR-224-5p*.

### 3.5. miR-224-5p Attenuates the Inflammatory Response via Targeting of ADAM17

The target genes of *miR-224-5p* were identified as a total of 28 target genes through prediction (Figure 4A). Among these genes, *ADAM17* was found because it is related to immune regulation. *miR-224-5p* mimic/inhibitor and their negative controls were transfected into GMECs. Results showed that the expression of the *ADAM17* gene was significantly decreased in GMECs treated with *miR-224-5p* mimic (*p* < 0.01, Figure 4B). Expression of the *ADAM17* gene was significantly increased in GMECs treated by *miR-224-5p* inhibitor (*p* < 0.01, Figure 4B), showing that *miR-224-5p* can negatively affect *ADAM17*.

The mRNA and protein levels of the *ADAM17* gene in the PCDNA3.1-*ADAM17* treatment group increased significantly (*p* < 0.01, Figure 5A,B). Three different siRNA of *ADAM17* could each significantly reduce the mRNA expression level of the *ADAM17* gene compared to the NC group (*p* < 0.01, Figure 5D); the results show that transfection of siRNA-*ADAM17*-443 will significantly reduce the expression of *ADAM17* mRNA and protein compared with the siRNA-NC group (*p* < 0.05, Figure 5E). At the same time, overexpression of *ADAM17* significantly increased the mRNA expression levels of IL-6 (*p* < 0.01, Figure 5C), IL-8 (*p* < 0.01, Figure 5C), and TNF-α (*p* < 0.01, Figure 5C). In contrast, interference with the expression of *ADAM17* significantly reduced the mRNA expression levels of TNF-α, IL-6, and IL-8 (*p* < 0.01, Figure 5F). Meanwhile, compared with the *miR-224-5p* inhibitor negative control and siRNA-NC groups, the expression of *ADAM17* and TNF-α genes in the *miR-224-5p* inhibitor and siRNA-NC groups were significantly increased (*p* < 0.01, Figure 6). The mRNA expressions of IL-6 and IL-8 were also increased (*p* < 0.05, Figure 6). Next, compared with the *miR-224-5p* inhibitor and siRNA-NC groups, the mRNA expressions of *ADAM17*, *TNF-α*, *IL-6* and *IL-8* in the *miR-224-5p* inhibitor and the siRNA group were significantly decreased (*p* < 0.01). Taken together, the data we have suggest that *miR-224-5p* down-regulates inflammatory response via targeting of *ADAM17*.

## 4. Discussion

Excess inflammation of the breast may contribute to pathological development of mastitis disease, which in turn causes adverse consequences for lactation performance and the physical health of dairy goats. In this study, we found that *LF* inhibited inflammatory response in GEMCs [7,8]. Furthermore, we propose that the miR-214-5p/LF/miR-224-5p/ADAM17 axis participates in the immune regulation of GEMCs.

It was found that *LF* gene expression was species-specific. In the primary structure, the homology of the *LF* gene between humans and bovines was 66%. There was also a big difference between human and mouse *LF*, and *LF* expression also had space–time specificity. In the mouse uterus, the *LF* gene was only expressed at 1–8 days of pregnancy [17]. Studies have found that expression of the *LF* gene in the uterus is the result of the combined action of estrogen and progesterone, and changes specifically with animal estrus [18]. The concentration of *LF* in human milk is high, especially in colostrum, which can reach up to 6~14 mg/mL. The concentration of *LF* in milk decreases with the prolongation of lactation to about 1 mg/mL. *LF* concentration in milk is relatively low, only 100~600 μg/mL [19]. The average concentration of *LF* in the milk samples of goats was 10–28 μg/mL at the peak and middle stages of lactation, while it increased by 3.2 times up to 107 ± 19 μg/mL at the end of lactation [20]. It was reported that LF was able to play an anti-inflammatory role. For example, a previous study indicated that LF may be involved in the transcriptional regulation of certain genes in the host inflammatory response, thereby inhibiting pro-inflammatory cytokines as transcription factors and regulators of the inflammatory process [21]. LF has strong anti-inflammatory activity and can regulate the inflammatory response of epithelial cells [9,22,23]. Consistent with this, our study also demonstrated that *LF* overexpression suppressed inflammatory response with decreasing *IL-6* and *IL-8*, as well as increasing *IL-10* mRNA levels. We predicted that *miR-214-5p* is targeted at the 3′UTR of *LF* mRNA through miRanda, Targetscan, DIANATOOLS, and other databases. Furthermore, a previous study has shown that miR-214 inhibits the expression of *LF* and promotes cell apoptosis in human mammary epithelial cells [24]. It has been perviously suggested that MiR-214 can also regulate the expression levels of intracellular inflammatory cytokines IL-6 and IL-1β in bovine mammary epithelial cells by targeting NFATc3 and TRAF3 [25]. In addition, *miR-214-5p* has been shown to inhibit inflammatory response by respectively targeting HOXA13 [26], PAK4 [27], and SIRT2 [28]. In the present study, the levels of inflammatory factors *IL-6* and *IL-8* were significantly increased in cells after overexpression of *miR-214-5p*. In parallel, the level of anti-inflammatory factor *IL-10* was significantly reduced. In contrast, inhibiting *miR-214-5p* can result in the opposite results in GMECs. Collectively, these data suggested that *miR-214-5p* directly targets *LF* and plays an important regulatory role in mammary immunity and health in goats.

To further confirm the improved regulatory network behind *LF*’s action, we mapped the *LF* regulatory gene network using miRNA-seq. By integrated analysis of these sequenced data, we found 22 differentially miRNAs directly regulated by *LF* overexpression. Among these differential miRNAs, *miR-224-5p* was reported to be associated with immune regulation. In mouse bronchial epithelial cells, overexpression of miR-224 can inhibit the expression of TLR2, significantly reduce the expression of IL-4, IL-5, and IL-17, and significantly increase the expression of IL-10, thereby inhibiting the inflammation of bronchial epithelial cells [29]. In this study, *LF* overexpression enhanced *miR-224-5p* mRNA level, which might be associated with improved anti-inflammatory capability. In line with our findings, it has been reported that the expression levels of *IL-6* and *TNF-α* in the LPS+ mimic NC group were significantly higher than those in the LPS+ miR-224 mimic group. The expression levels of *IL-6* and *TNF-α* in the LPS+ miR-224 inhibitor group were also significantly increased compared with the LPS+ inhibitor NC group [30]. Altogether, *LF* exhibited an anti-inflammatory effect in GMECs, partly via targeting inhibition of *miR-224-5p*.

Next, we investigated the regulation of *miR-224-5p* on inflammatory factors in GMECs and tried to find target genes of *miR-224-5p*. Utilizing target gene prediction online software, the unknown target gene *ADAM17* of *miR-224-5p* was screened. Of note, the *ADAM17* gene was one of the candidate target genes for *miR-224-5p* and may be related to immune regulation. *ADAM17* played an important role in the regulation of inflammation, which was supported by previous findings showing that *ADAM17* expression in endothelial cells affected renal inflammation and modulated renal function and histology in an obese prediabetic mouse model [31]. Furthermore, miR-224 can inhibit the growth and invasion of oral squamous cell carcinoma (OSCC) cells by targeting *ADAM17* expression [32]. In this work, *miR-224-5p* up-regulation contributed to anti-inflammation ability by decreasing the expression of the ADAM17 gene in GMECs. Furthermore, after knocking down *ADAM17*, the expression of TNF-α, IL-6, and IL-8 in cells decreased. Co-transfection assays showed that *miR-224-5p* down-regulates the expression of goat inflammatory response via targeting *ADAM17*. These data demonstrate that *ADAM17* is required for *miR-224-5p* in inhibiting inflammatory response. Altogether, this study demonstrates that the miR-214-5p/LF/miR-224-5p/ADAM17 axis can effectively regulate the expression of cytokines related to cellular inflammation.

## 5. Conclusions

This study has demonstrated that *miR-214-5p*/*LF* target *miR-224-5p*/*ADAM17* and negatively affect the immune factors’ levels. Furthermore, *miR-224-5p* regulates transcription of the IL-6, IL-8, and TNF-α genes through *ADAM17* expression. In summary, this study contributes to the exploration of the anti-inflammatory effects and molecular mechanisms of *LF*, providing a theoretical reference for the prevention and cure of inflammation by manipulating bioactive milk proteins in ruminants.

## Figures and Tables

**Figure 1 animals-13-02835-f001:**
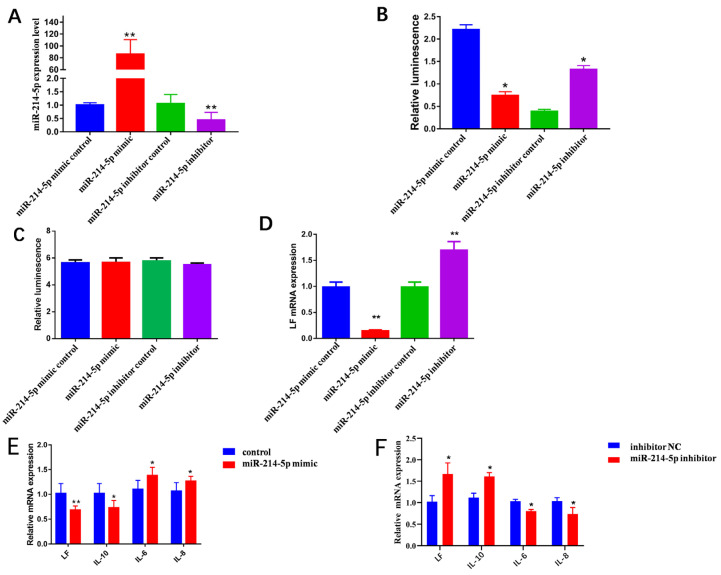
The effects of miR-214 on *LF* and inflammatory factors. (**A**) Quantitative detection of miR-214 from the transfection efficiency of miRNA mimic after the *miR-214-5p* mimic/control and its inhibitor/control synthesized by the company were diluted according to standards and transfected into GMECs and the cell miRNA was extracted according to the instructions of the kit. (**B**) The constructed *LF* dual-luciferase vector was co-transfected with *miR-214-5p* mimic/control and *miR-214-5p* inhibitor/control into GMECs. After 48 h, the cells were collected to verify the target relationship between *miR-214-5p* and *LF* by detecting the dual luciferase activity. (**C**) The constructed mutant double-fluorescent plasmid was co-transfected into GMECs with 50 nM *miR-214-5p* mimic/control and 100 nM *miR-214-5p* inhibitor/control, respectively. After 48 h of culture, the cells were collected to detect the luciferase activity to detect the binding site of *LF* and *miR-214-5p*. (**D**) GMECs were cultured in 12-well plates and 6-well plates, *miR-214-5p* mimic (50 nM)/inhibitor (100 nM) transfection was performed when the confluence degree of cells reached about 70%, and the control group was transfected with the control. After 24 h of culture, 12-well cells were collected. The total RNA of the cells was extracted by the Trizol method, and RNA inversion was performed. Quantitative fluorescence PCR was performed to detect *LF* mRNA levels after overexpression of *miR-214-5p*. (**E**) After the cells were transfected with miR-214 mimic, the expression levels of inflammatory factors and *LF* in cells were quantitatively detected. (**F**) After the cells were transfected with miR-214 inhibitor, the expression levels of inflammatory factors and *LF* in cells were quantitatively detected. * Significant difference (*p* < 0.05). ** Extremely significant difference (*p* < 0.01).

**Figure 2 animals-13-02835-f002:**
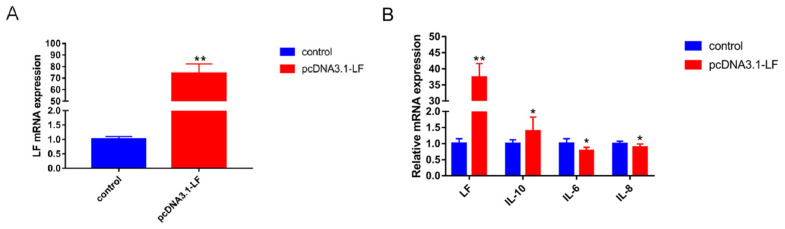
*LF* regulates the expression of cellular inflammatory factors. (**A**) The *LF* overexpression vector was transfected into GMECs, and the cells were collected after 24 h of culture to quantitatively measure the mRNA level of *LF*. (**B**) All GMECs were transfected with the *LF* overexpression vector, and the expression of inflammatory factors was quantitatively detected. * Significant difference (*p* < 0.05). ** Extremely significant difference (*p* < 0.01).

**Figure 3 animals-13-02835-f003:**
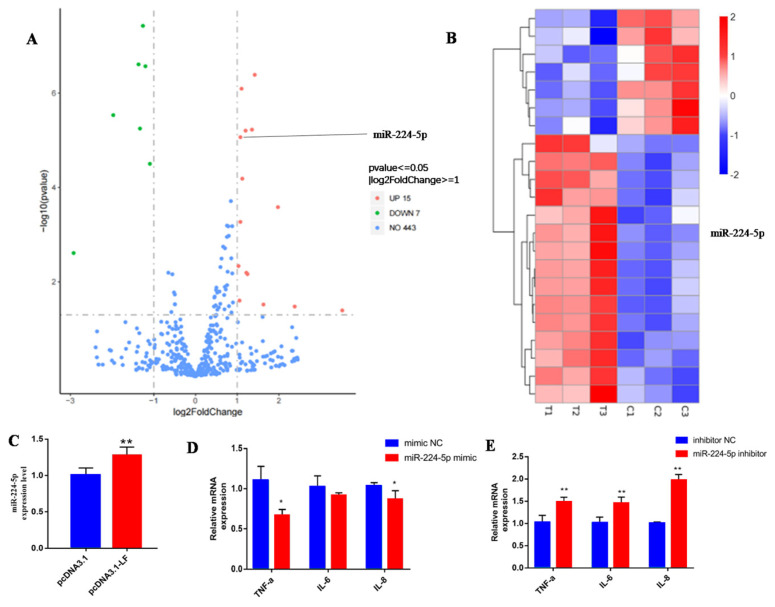
The differential miRNA *miR-224-5p* was screened through overexpression of *LF* and its effects on cellular inflammatory factors. (**A**) The *LF* overexpression vector was transfected into GMECs, and a volcano map of miRNA differential expression was obtained by high-throughput sequencing analysis. (**B**) The *LF* overexpression vector was transfected into GMECs, and a heat map of miRNA differential expression was obtained by high-throughput sequencing analysis. (**C**) Validation of *miR-224-5p* fluorescence quantitative PCR results. (**D**) GMECs were cultured until cell density reached 60–80%; *miR-224-5p* mimic NC, mimic, inhibitor NC, and inhibitor were diluted in DMEM/F12, mixed with transfection reagent, and added to the cell culture medium to treat the cells. The effect of up-regulated *miR-224-5p* on inflammation factors was then analyzed. (**E**) GMECs were cultured until cell density reached 60–80%; *miR-224-5p* mimic NC, mimic, inhibitor NC, and inhibitor were diluted in DMEM/F12, mixed with transfection reagent, and added to the cell culture medium to treat the cells. The effect of down-regulated *miR-224-5p* on inflammation factors was then analyzed. * Significant difference (*p* < 0.05). ** Extremely significant difference (*p* < 0.01).

**Figure 4 animals-13-02835-f004:**
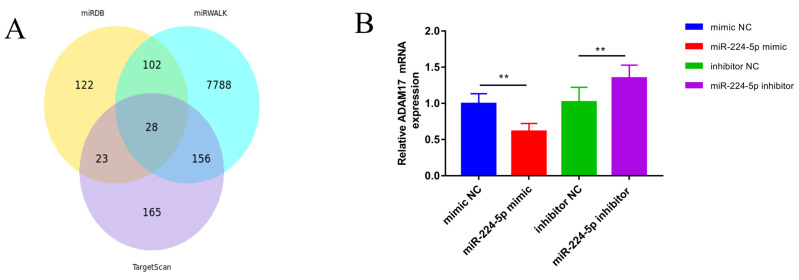
Verification of the target relationship between *miR-224-5p* and *ADAM17*. (**A**) Venn diagram of *miR-224-5p* target gene prediction by miRDB, miRWALK, and TargetScan. (**B**) GMECs were cultured until the cell density reached 60–80%; *miR-224-5p* mimic NC, mimic, inhibitor NC, and inhibitor were diluted in DMEM/F12, mixed with transfection reagent, and added to the cell culture medium to treat the GMECs. The effect of *miR-224-5p* on the mRNA level for the candidate target gene *ADAM17* was then analyzed. ** Extremely significant difference (*p* < 0.01).

**Figure 5 animals-13-02835-f005:**
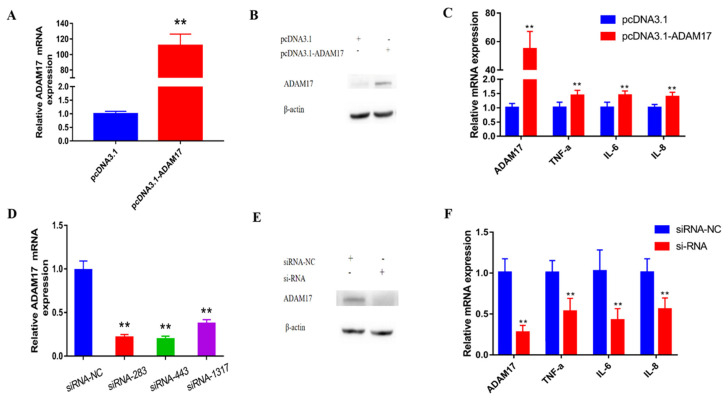
*ADAM17* regulates the expression of cellular inflammatory factors. (**A**) The pcDNA3.1 high-purity plasmid and pcDNA3.1-*ADAM17* high-purity plasmid were transfected into 60–80% mammary epithelial cells, respectively, and the expression of *ADAM17* mRNA was detected after the overexpression. (**B**) The pcDNA3.1 empty vector and pcDNA3.1-*ADAM17* vector were transfected into cells for 48 h, and the total protein was extracted for Western blotting to detect the change in *ADAM17* gene protein level. (**C**) After overexpression of *ADAM17*, RNA was extracted from the cells after 24 h of culture, and the mRNA expression levels of *ADAM17*, TNF-α, IL-6, and IL-8 were detected by fluorescence quantitative technology to show the effects of *ADAM17* overexpression on inflammation factors. (**D**) After completion of siRNA synthesis, the samples were centrifuged and diluted with RNA-free water and stored at −20 °C. GMECs were cultured, and, when the cell density reached 60% to 80%, siRNAs were diluted in DMEM/F12 and transfected at a final concentration of 100 nM. After the three interfering RNAs were transfected into cells, the interference efficiency on the mRNA level of *ADAM17* was detected. (**E**) The interference efficiency of *ADAM17* at the protein level was detected after transfecting three interfering RNAs into cells as described above. (**F**) After interference on *ADAM17*, RNA was extracted from the cells after 24 h of culture, and the mRNA expression levels of *ADAM17*, TNF-α, IL-6 and IL-8 were detected by fluorescence quantitative technology to show the effects of *ADAM17* interference on inflammation factors. ** Extremely significant difference (*p* < 0.01).

**Figure 6 animals-13-02835-f006:**
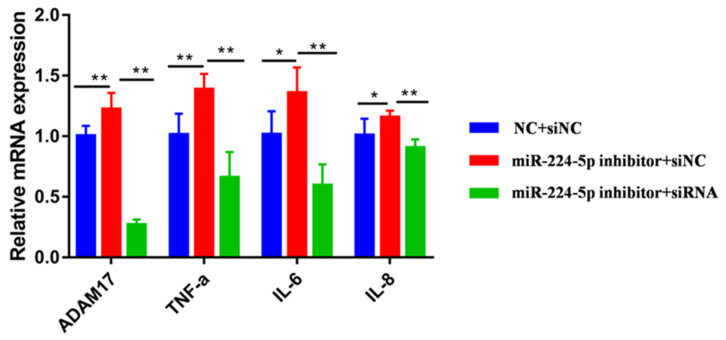
Effects of *miR-224-5p* inhibitor and siRNA co-transfection on *ADAM17* and inflammation factors. * Significant difference (*p* < 0.05). ** Extremely significant difference (*p* < 0.01).

**Table 1 animals-13-02835-t001:** Primers Used for Quantitative Real-Time PCR of Genes.

Gene	Primer Sequences (5′~3′)
*LF*	F: TCTGGCTGCCCCGAGGAAAAACGTT
	R: TCCTGCAGGCACTTGAAGGCACCAG
*miR-214-5p*	F: ACAGCAGGCACAGACAGGCAGT
	R: TCTCAGAAGCTAAACAGGGTCG
*ADAM17*	F: GGCGCGGGAGGAATAAGAAG
	R: TGTCCAGGAAATCAGAGAGCC
IL6	F: AGATATACCTGGACTTCCT
	R: TGTTCTGATACTGCTCTG
IL8	F: AAGCTGGCTGTTGCTCTCTTG
	R: TGTTCTGATACTGCTCTG
IL10	F: CATGGGCCTGACATCAAGGA
	R: CTCTTGTTTTCGCAGGGCAG
TNF-α	F: TGGTTCAGACACTCAGGT
	R: CGCTGATGTTGGCTACAA
si*ADAM17*	F: CCGCUUUGGAGACUAAUUATT
	R: UAAUUAGUCUCCAAAGCGGTT
MRPL39	F: AGGTTCTCTTTTGTTGGCATCC
	R: TTGGTCAGAGCCCCAGAAGT
UXT	F: TGTGGCCCTTGGATATGGTT
	R: GTGTCTGGGACCACTGTGTCAA

## Data Availability

Data is contained within the article.

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
