# Peer review of "The miR-214-5p/Lactoferrin/miR-224-5p/ADAM17 Axis Is Involved in Goat Mammary Epithelial Cells’ Immune Regulation"

_animals, 2023, doi:10.3390/ani13182835_

Round 1

Reviewer 1 Report

Summary: The manuscript of Pang et al. “The MiR-214-5p/LF/MiR-224-5p/ADAM17 Axis is involved into 2 GEMCs Immune Regulation” investigates the molecular mechanisms of Lacroferrin regulation in goat mammary epithelial cells.

General comments:

The topic of the manuscript is interesting and relevant in the field, but the manuscript need a deep English editing and the experimental design presents important bias.

Specific comments

Abstract:

From the abstract it is not easy to understand the experimental design and aim of the study.

Introduction:

The introduction should completely re-organized. In the present form the introduction is very confused; there are different arguments but the connection among them it is not clear. The authors should include a brief description of miRNA and their activity mechanism in the introduction section.

Several sentences need the relative reference.

L39: delete “in goat mammary epithelial primary cells”, it is a repetition.

L42: what does it means “milk cells”?

L48: substitute “normal” with “mature”.

L49: what the authors means by “natural immunity”? innate immunity?

L51: only intracellular bacteria?

L54-58: the sentence is not clear.

L64: substitute “affect” with “affects”.

The authors should briefly described their experimental model and aims in the introduction.

Materials and Methods:

Mammary tissue collection and cell culture

The authors should indicate the age and lactation phase of the sampled animals. From how many animals the authors collected the samples? GEMC were pooled from different animals or they come from a single animal?

The differential adhesion collection it is not sufficient to guarantee a high purity % of GEMC. Did the authors check the purity of their cells? A cell sorting approach should be more appropriated. This represent an important bias, because the authors cannot be sure that the pathway they described is specific for GEMC. The pathway could be altered by the presence of other type of cells: endothelial cells, fibroblast, adipocytes….

L93-94: the sentence is not clear. More cultural detail should be present: medium? Use of FBS, anibiotics, other reagents, concentration of plated cells/ml of medium, incubation conditions.

miRNA seq

All the acronyms should be presented in the extended form the first time it is mentioned in the text (GEMC, FAPs, BM, ADM, …….).

Vector construction

It is not clear if the author produced a fusion protein LF-ADAM17 or two constructs (LF and ADAM17). A figure with the scheme of the different constructs in the supplementary material could help the reader.

The experimental design it is not clear: which are the experimental groups? How many replicates per treatment? Which controls were used?

Did the authors check the basal expression of miRNAs, LF and ADAM17 in GEMC?

GEMCs cell transfection

 The cell transfection condition should be deeply described.

RNA Extraction and Quantitative Real-Time PCR (qRT-PCR)

How mRNA was retrotranscribed? The author declare that they used the relative quantification (2-DDCt). This method need the definition of a calibrator, but the authors didn’t mention which sample was used as calibrator. May be they used 2-DCt method?

L153: substitute “transferation” with “transfer”.

WB

The author should add several detail of the western blot analysis: total protein amount loaded per lane, temperature and duration of the incubations, primary antibodies used, …..

Luciferase assay

The author should add several detail of the luciferase assay: type of cells (GEMC?), concentration of cells/ml or well, luciferase vectors used (promoter of ADAM17, promoter of LF?).

L 167-169: the sentence is not clear.

L172: the authors means “triplicate”? for each condition?

L176: what does it means: “using control to control”?

Statistical analysis

Did the authors perform a normality test on each parameter? Student’s t-test and Anova test can be applied to normally distributed parameters…..

Results:

L188-190: delete

L202: what is “psi-MUT-LF”? It is not mentioned in Materials and Methods section.

L209-210: move the sentence to the Discussion section.

L228: miRNA seq was made in which samples?

L236: why in paragraph 3.3 the author evaluated IL-6 and IL-8, whereas in 3.4 they evaluated TNFa and IL-8?

L239-40: move the sentence to the Discussion section. The present study do not exclude the activity also of other pathway in the regulation of LF and inflammation in mammary cells.

L250: compared to?

L255: the sentence is not clear.

L263-65: move the sentence to the Discussion section.

Figures

Please add to the graphs the units of measures. Some figure has a very low quality and definition and it is difficult to read. Using the calibrator in each graph there should be the calibrator=1, but it is not true in all the graphs.

Figure 3B: please indicate the transfected cells and the control cells.

Discussion:

In the Discussion section several references are missed. The first part of the Discussion section seems an introduction….

L339-42: this paragraph is very difficult to follow; there are a lot of concepts that are not linked one each other. It is not correct to add the transcription factors among the factors that influence the regulation of LF, they are the molecular effectors of the regulation.

L375-82: the author should discuss about the opposite result in different studies.

L389-401: the comparisons among different species are not clear.

The authors should add a figure that represent the propose pathway.

Summary: The manuscript of Pang et al. “The MiR-214-5p/LF/MiR-224-5p/ADAM17 Axis is involved into 2 GEMCs Immune Regulation” investigates the molecular mechanisms of Lacroferrin regulation in goat mammary epithelial cells.

General comments:

The topic of the manuscript is interesting and relevant in the field, but the manuscript need a deep English editing and the experimental design presents important bias.

Specific comments

Abstract:

From the abstract it is not easy to understand the experimental design and aim of the study.

Introduction:

The introduction should completely re-organized. In the present form the introduction is very confused; there are different arguments but the connection among them it is not clear. The authors should include a brief description of miRNA and their activity mechanism in the introduction section.

Several sentences need the relative reference.

L39: delete “in goat mammary epithelial primary cells”, it is a repetition.

L42: what does it means “milk cells”?

L48: substitute “normal” with “mature”.

L49: what the authors means by “natural immunity”? innate immunity?

L51: only intracellular bacteria?

L54-58: the sentence is not clear.

L64: substitute “affect” with “affects”.

The authors should briefly described their experimental model and aims in the introduction.

Materials and Methods:

Mammary tissue collection and cell culture

The authors should indicate the age and lactation phase of the sampled animals. From how many animals the authors collected the samples? GEMC were pooled from different animals or they come from a single animal?

The differential adhesion collection it is not sufficient to guarantee a high purity % of GEMC. Did the authors check the purity of their cells? A cell sorting approach should be more appropriated. This represent an important bias, because the authors cannot be sure that the pathway they described is specific for GEMC. The pathway could be altered by the presence of other type of cells: endothelial cells, fibroblast, adipocytes….

L93-94: the sentence is not clear. More cultural detail should be present: medium? Use of FBS, anibiotics, other reagents, concentration of plated cells/ml of medium, incubation conditions.

miRNA seq

All the acronyms should be presented in the extended form the first time it is mentioned in the text (GEMC, FAPs, BM, ADM, …….).

Vector construction

It is not clear if the author produced a fusion protein LF-ADAM17 or two constructs (LF and ADAM17). A figure with the scheme of the different constructs in the supplementary material could help the reader.

The experimental design it is not clear: which are the experimental groups? How many replicates per treatment? Which controls were used?

Did the authors check the basal expression of miRNAs, LF and ADAM17 in GEMC?

GEMCs cell transfection

 The cell transfection condition should be deeply described.

RNA Extraction and Quantitative Real-Time PCR (qRT-PCR)

How mRNA was retrotranscribed? The author declare that they used the relative quantification (2-DDCt). This method need the definition of a calibrator, but the authors didn’t mention which sample was used as calibrator. May be they used 2-DCt method?

L153: substitute “transferation” with “transfer”.

WB

The author should add several detail of the western blot analysis: total protein amount loaded per lane, temperature and duration of the incubations, primary antibodies used, …..

Luciferase assay

The author should add several detail of the luciferase assay: type of cells (GEMC?), concentration of cells/ml or well, luciferase vectors used (promoter of ADAM17, promoter of LF?).

L 167-169: the sentence is not clear.

L172: the authors means “triplicate”? for each condition?

L176: what does it means: “using control to control”?

Statistical analysis

Did the authors perform a normality test on each parameter? Student’s t-test and Anova test can be applied to normally distributed parameters…..

Results:

L188-190: delete

L202: what is “psi-MUT-LF”? It is not mentioned in Materials and Methods section.

L209-210: move the sentence to the Discussion section.

L228: miRNA seq was made in which samples?

L236: why in paragraph 3.3 the author evaluated IL-6 and IL-8, whereas in 3.4 they evaluated TNFa and IL-8?

L239-40: move the sentence to the Discussion section. The present study do not exclude the activity also of other pathway in the regulation of LF and inflammation in mammary cells.

L250: compared to?

L255: the sentence is not clear.

L263-65: move the sentence to the Discussion section.

Figures

Please add to the graphs the units of measures. Some figure has a very low quality and definition and it is difficult to read. Using the calibrator in each graph there should be the calibrator=1, but it is not true in all the graphs.

Figure 3B: please indicate the transfected cells and the control cells.

Discussion:

In the Discussion section several references are missed. The first part of the Discussion section seems an introduction….

L339-42: this paragraph is very difficult to follow; there are a lot of concepts that are not linked one each other. It is not correct to add the transcription factors among the factors that influence the regulation of LF, they are the molecular effectors of the regulation.

L375-82: the author should discuss about the opposite result in different studies.

L389-401: the comparisons among different species are not clear.

The authors should add a figure that represent the propose pathway.

Author Response

Thanks for your detailed suggestions and comments, we have revised the manuscript and uploaded a detailed reply file.

Reviewer 2 Report

animals-2525060-peer-review-v1

The study investigated the miR-214-5p/LF/miR-224-5p/ADAM17 axis in the immune regulation of goat epithelial mammary cells (GEMCs). Please specify the abbreviation in the title and the text.  

Overall, the paper is confusing in writing, and the hypothesis and purpose of the study are not clearly and concisely presented.

All research components are present and clearly stated in the materials and methods section.

Take out the first part of the introduction from lanes 34 to 46 or better explain the link of mastitis with LF.

Is LF the aim or ADAM17 or mastitis?

Delete the entire sentence from lanes 188 to 190.

The results, therefore, are not so logically presented and by the significance of findings. Even statements and conclusions are not supported by data and are linked to goals.

References are appropriate to the manuscript type.

Overall, the paper is confusing in writing, and the hypothesis and purpose of the study are not clearly and concisely presented.

Author Response

Thanks for your suggestions and comments, we have revised the manuscript and uploaded a detailed reply file.

Reviewer 3 Report

Dear, authors,

This paper is generally well written and structured. The authors report that this study investigated the anti-inflammatory effects and molecular mechanisms of LF to better understand how lactoferrin affects the immune system and to provide a reference for further study of lactoferrin and its application in the future. Therefore, I am accepting this paper in its present form.

Author Response

Thanks for your affirmation, we have further improved the manuscript on this basis.

Reviewer 4 Report

I have reviewed the paper entitled " The MiR-214-5p/LF/MiR-224-5p/ADAM17 Axis is involved into GEMCs Immune Regulation ". The manuscript identified the role network of lactoferrin in breast immunity through miRNA-sequencing and online prediction of target groups. This work is meaningful and interesting. Although the manuscript describes an interesting opinion, results could be more clearly presented and relevant aspects could be discussed, in more detail.

Minor:
1. Line 22:
The comma after the wordsystem should be removed, and the determiner appears to be missing before the subsequent word “reference”. There are also a few grammatical errors in other parts of the article, please pay attention to correction
2.
The Materials and Methods” section could add more detail.

3.The high-throughput sequencing component does not seem to be mentioned in the method. Maybe because it 's done by the company or someone else?
4. Line 154: Vector Construction. Many experimental procedures are missing in material and methods: Generation of stable transfectants overexpressing LTF and ADAM17. Meanwhile, this has not been presented in the results.

5. In figure 2 and figure 3, since multiple genes have been screened out, are only miR-224-5p and ADAM17 related to immunity? Have other genes with similar functions also been screened out?
6. Some part of this article have sentences may be wordy, consider changing the wording.
7. In general, the quality of writing can be improved to be satisfied. Suggest to use an English native speaker or Language Editing service to modify the language and to improve its quality.

Minor editing of English language required

Author Response

Thanks for your suggestions and comments, we have made extensive revisions to the manuscript to perfect it. We have also uploaded a detailed response for your review.

Round 2

Reviewer 1 Report

The new version of the manuscript is improved, but it still need other improvement. Some of the requred reviwer modifications have not been done.

The main problem remain the method of GEMC isolation and the evaluation of their purity. This element is crucial for the conclusions of the study.

The English should be further edited.

Reviewer 2 Report

The authors make a reasonable effort to improve the manuscript following advice. For me can be accepted in the present revised form.